# Effects of Task Interference on Kinematics and Dual-Task Cost of Running in Early Childhood

**DOI:** 10.3390/s24051534

**Published:** 2024-02-27

**Authors:** Panchao Zhao, Kai Ma, Zhongqiu Ji, Guiping Jiang

**Affiliations:** 1Department of Physical Education, China University of Geosciences (Beijing), Beijing 100083, China; makai@cugb.edu.cn; 2College of P.E. and Sports, Beijing Normal University, Beijing 100875, China; jizhongqiu61@bnu.edu.cn (Z.J.); jiang_guiping0401@126.com (G.J.)

**Keywords:** early childhood, running, interference, kinematics, dual-task cost

## Abstract

Children aged 3–8 are in a critical period for motor development and postural control. Running is a basic motor skill that children need to master in early childhood. While running, children are prone to dangerous events such as falls. This study investigates the kinematic characteristics of running by children associated with different interference tasks, i.e., normalized running, cognitive dual-tasks, and obstacle crossing tasks, and provides a theoretical foundation for the interference mechanism of children’s dynamic postural control and for screening of motor disorders. Two hundred children aged 3–8 were recruited. The BTS Bioengineering infrared motion capture system was used to collect spatiotemporal and kinematic running data under three tasks. Repeated measures of variance analysis were used to compare the effects of different interference tasks and ages on children’s running signs. The main and interaction effect tests were compared by the Bonferroni method. The results and conclusions are as follows: (1) Running characteristics of early childhood are influenced by interference tasks and age. With interference tasks, the overall characteristics of running by children aged 3–8 showed an increasing trend in running cycle time and a decreasing trend in stride length, step length, cadence, and speed. (2) Both cognitive and obstacle crossing tasks had costs, and cognitive task costs were greater than obstacle crossing costs. Children adopted a “task first” running strategy with different interference tasks. When facing cognitive tasks, their overall joint motion decreased, and they reduced joint motions to promote task completion. When facing obstacle crossing tasks, because of the characteristics of the task itself, children increased joint motions to cope with interference. (3) In terms of age, the running characteristics showed a nonlinear development trend in various indicators, with a degree of recurrence and high variability in adjacent age groups. (4) The dual-task interference paradigm of “postural-cognition” can be used as a motor intervention tool to promote the development of basic motor skills in early childhood.

## 1. Introduction

Running is a basic motor skill that children need to master in early childhood, and the age of 2.5–8 is a critical period in which to learn running skills [1]. Mature running patterns are automated and do not require excessive attention. However, in early childhood, running patterns belong to the developmental stage and require more attention to maintain balance and complete motors. Running is easily disrupted by competing tasks and instability factors during this period [2,3], and attention must be allocated to motor, cognitive, and sensory stimulation [4].

Research on the effects of interference on a child’s postural control mainly includes paradigms such as dual-task interference and obstacle avoidance strategies. In a dual-task study of “gait and handling boxes”, Hung et al. found that children aged 4–6 have lower coordination abilities compared to baseline levels, and their ability to perform dual-tasks is still in the developmental stage [5]. The complexity of tasks can affect the stability of a child’s gait. Abbruzzese et al. compared the effects of different levels of complexity of motor tasks on a child’s gait. They found that children experienced a decrease in gait speed and cadence, and an increase in double support time for all tasks; complex tasks had a greater effect on gait stability [6]. Research on the dual-task gait of children showed that the cost of performing the second task while walking is a decrease in gait speed and cadence and an increase in gait cycle time.

Human attention resources are limited. When the demand for parallel tasks exceeds the processing ability of the executor, interference effects can occur [7]. Dual-tasks and external interference tasks can affect a child’s running characteristics and stability, which are related to age, task difficulty, and attention resources. At present, research on the dual tasks of postural control in early childhood focuses mainly on gait, but running is prone to dangerous events such as falls. Therefore, we used the dual-task paradigm of “cognition dual-task” and obstacle crossing task to study the effects of different interference tasks on children’s running characteristics. Our goal was to enrich the theoretical content and framework of children’s postural control, to enable timely screening of children with motor disorders, and to provide a reference for addressing these conditions.

## 2. Materials and Methods

### 2.1. Design

Experiment data on running kinematics for 3–8-year-old children were used for repeated measures of two-way analysis of variance (ANOVA). The intra-group independent variables were interference tasks, and the inter-group independent variable was age. The main effect refers to the influence of a single factor on the kinematic parameters of running, while the interaction effect refers to the influence of two factors on the kinematic parameters at the same time, which occurs in one of the age group or an interference group.

### 2.2. Time and Location

The experiment began in May 2021 and lasted for two months. The study was conducted at a public kindergarten and a primary school in Heibei province.

### 2.3. Participants

Two hundred children aged 3–8 years were recruited by inclusion criteria. Before data collection, written informed consent was obtained from the parents of participants. The study was conducted in accordance with the Code of Ethics of the World Medical Association (Declaration of Helsinki), and the Ethics Committee of the Psychology Department at Beijing Normal University approved the protocol (201910210061). According to the exclusion and shedding criteria, 180 participants’ data were enrolled.

Inclusion criteria were as follows: (1) age 3–8; (2) normal cognitive function and good comprehension skills; (3) normal motor ability and the capacity to independently complete running tasks.; and (4) guardian fully understands the research content and voluntarily signs an informed consent form.

Exclusion criteria were as follows: (1) developmental and skeletal muscle coordination disorder; (2) cognitive impairment; and (3) inability to complete required actions.

Shedding and rejection criteria were the following: (1) participant unable to cooperate with the research; (2) participant requests to withdraw from the study midway; and (3) participant with incomplete data capture.

Five testers were required to collect data. Two were responsible for pasting the markers to the subjects, one was at the start position of the runway to tell the subjects when to start, one was at the end position to protect the children’s safety braking, and one was responsible for instruments and equipment, mainly computer operation and data collection. All children’s data were completed by the same five testers. Before the formal test, they were trained in the operation of the above tasks to ensure the accuracy of the test results.

### 2.4. Method

#### 2.4.1. Experimental Equipment

Bioengineering Technology and Motion Capture System SMART DX 700 (Milano, Italy) was used to collect running kinematic data. Data were acquired at 100 Hz and the meanings are shown in Table 1. Markers were applied according to the Davis model, which is typically used for assessing gait performance in children [8], with positions on the midpoint of the clavicle, acromion, anterior superior iliac spines, posterior superior iliac spines, greater trochanter of femur, midpoint of the thigh, lateral femoral epicondyles, fibular head, midpoint of the shank, lateral malleoli, heel, and the fifth metatarsal head.

#### 2.4.2. Procedures and Protocol

The test was conducted in an empty classroom of the kindergarten. The room was 15 m × 8 m × 4 m, the runway was set at the center of the field, and 8 infrared cameras were placed in a circular shape around the runway. Before the test, a 3D calibration framework was used to scan the test field to eliminate interference from other reflective markers. The subjects wore clean and tight test clothing and stick markers. Participants warmed up for five minutes and performed three running tests barefoot at their chosen speed. We ensured that only one child was in the room during the test. After one child completed the tests, we proceed to the next child’s test. Two tests were conducted for each running type and there was a 1-min rest after both tests. After normal running, the cognitive tasks and obstacle crossing tasks were performed randomly. The specific testing requirements were the following:

A normalized running test with a test trail of 8 m. After hearing the “start” command, the participants ran from the starting position to the ending position.

A cognitive dual-task running test. While completing the normalized running test, participants were required to perform another cognitive task conducted in the form of questioning. The cognitive task required participants to answer questions while running, and to not stop running as much as possible. Before the test, learning materials of children in different ages were consulted and discussed with the class teachers to understand the cognitive development level of children and to formulate cognitive questions for children of different ages.

An obstacle crossing running test in which obstacles were set at a distance of 3 m from the starting point, at a height of 5% of each child’s height [9]. Participants were required to cross obstacles during running (testing site images are shown in Appendix A).

#### 2.4.3. Data Processing

BTS Bioengineering clinic gait analysis software (SMART DX 700) was used to calculate running spatiotemporal parameters and to collect parameters such as height, weight, pelvic width, knee width, and ankle width of the participants for modeling. The running kinematic parameters selected included running cycle time, stance phase percentage, swing phase percentage, stride length, step length, cadence, and speed. The formula for the calculation of dual-task cost (DTC) of stride length and speed was the following [10]:$$DTC=\frac{Single\;task\;performance\;-\;Dual\;-\;task\;performance}{Single\;task\;performance}\times 100\%$$

### 2.5. Data Processing

Statistical analysis of indicators was performed with SPSS 23.0 software. The results were averaged across all participants and the data were described by mean ± standard deviation ($\bar{x}$ ± s). Two-way ANOVA with repeated measures was used for comparison between groups and Bonferroni multiple comparisons were used for post-hoc tests, with the significance level taken as α = 0.05.

## 3. Results

### 3.1. Basic Information of Participants

The average age of the participants was 6.05 ± 1.64 years, their average height was 118.31 ± 11.30 cm, average weight was 23.73 ± 6.83 kg, and average BMI was 16.67 ± 2.54 kg/m^2^. Table 2 shows basic information about the participants.

### 3.2. Percent of Running Cycle

Figure 1 shows the effect of tasks and age on running cycle. The red box represents the main effect of task, and the running cycle of cognitive and obstacle crossing tasks was greater than the normalized running (*p* < 0.05). The stance phase percentage of cognitive tasks was higher than normalized and obstacle crossing tasks (*p* < 0.05), whereas the swing phase percentage was lower than normalized and obstacle crossing tasks (*p* < 0.05). Age characteristics of the running cycle time showed a nonlinear increasing trend under normalized and cognitive tasks, with significant differences observed in the 3-year-old, 5-year-old, and 8-year-old groups. In obstacle crossing tasks, the swing phase percentage showed a nonlinear increasing trend with age, and the 5-year-old, 7-year-old, and 8-year-old groups were larger than the 3-year-old group (*p* < 0.05).

### 3.3. Spatial Characteristics of Running

Spatial parameters of running showed the difference of main effect, without any interactive effect. Figure 2 shows the main effect of task on spatial parameters of running. The stride length, step length, cadence, and speed of cognitive tasks were all lower than normalized and obstacle crossing tasks (*p* < 0.05), whereas the baseline speed was higher for cognitive and obstacle crossing tasks (*p* < 0.05).

Table 3 shows the results of multiple comparisons. Overall, as age increased, stride length and step length increased, whereas cadence and speed decreased. There was a significant difference between the 7–8-year-old group and the 3–4-year-old group (*p* < 0.05).

### 3.4. Lower Limb Joint Kinematic Characteristics

Figure 3 shows the main effect of task on lower limb joint angle. From the overall characteristics, the joint angle of cognitive tasks showed the lowest value (*p* < 0.05), whereas the obstacle crossing tasks showed the highest value (*p* < 0.05).

Table 4 shows the multiple comparison results. As age increased, the range of motion of the hip, knee, and ankle joints, hip landing angle, hip pushing angle, and knee landing angle showed a nonlinear decreasing trend, and the knee pushing angle and ankle pushing angle showed a nonlinear increasing trend. There was an interactive effect on the ankle landing angle. In terms of task characteristics, the landing angle of obstacle crossing was the largest. The normalized landing angle of the 4–5-year-old group was greater than the cognitive angle (*p* < 0.05), whereas the normalized landing angle of the other groups was smaller than the cognitive angle (*p* < 0.05). As age increased, the landing angle showed an increasing trend. The angular acceleration of the three angles showed a decreasing trend. Overall, the significant differences in age characteristics of joint angles were more evident in cognitive dual-task, with feature differences observed between the 3–4 and 5–8 age groups (*p* < 0.05).

### 3.5. Dual-Task Cost

The task cost is the main effect of task and age; there was no interaction effect. Table 5 shows multiple comparison results. Figure 4 shows the age trend of dual-task costs, with children’s speed and stride DTC decreasing with age. The cognitive task costs of running speed and stride length were both higher than obstacle crossing tasks (*p* < 0.05). In obstacle crossing tasks, the speed cost of the 8-year-old group was lower than the 3-year-old group (*p* < 0.05), whereas the stride length cost of the 7-year-old group was lower than the 3-year-old group (*p* < 0.05).

## 4. Discussion

Ages 3–8 are the crucial period for childhood development of basic motor skills, and these basic motor skills have a foundational function in the later learning of complex motor skills [11]. Running is a basic motor skill that children need to master. Running mode is not mature in early childhood. When another parallel task is performed simultaneously with running, more requirements are placed on a child’s postural control system.

The spatiotemporal parameters of running describe the speed of completion and reflect the most basic motor characteristics [12]. With cognitive and obstacle crossing tasks, children’s running cycle time increased. For cognitive tasks, stance percentage increased and swing percentage decreased, and for obstacle crossing tasks, stance percentage decreased and swing percentage increased. The running cycle time reflects the speed of the running process. With interference tasks, a child’s running process is disrupted, resulting in more running cycle time. Generally speaking, mature running patterns have less stance time and more swing time [13]. With cognitive tasks, children experience an increase in stance time and a decrease in swing time, resulting in an unstable running pattern. However, because of the special nature of the task, obstacle crossing causes a decrease in stance time and an increase in swing time.

The spatial parameters mainly include stride length, step length, cadence, and speed. Two types of tasks have an impact on the spatial parameters of running, with stride length, step length, cadence, and speed all being smaller than standard running. Cognitive tasks have a greater impact on the spatial parameters of gait. With cognitive tasks, children’s running spatiotemporal parameters exhibit a minimum value. Whittal studied the effect of cognitive tasks (including question answering and memory tasks) on the running speed of girls aged 2.5 to 10 and adult women. The study showed that both cognitive tasks reduced walking speed in all age groups, with a significant decrease for children aged 4 and 6 [14].

Kinematic parameters are also important indicators of gait characteristics. Joint angle represents the amplitude of joint motion, and angular velocity is the rate of change in joint angle. The greater the change in joint angle per unit time, the faster the joint movement [15]. There is a difference in the joint angle between cognitive and obstacle crossing tasks, with the cognitive task having the minimum joint angle and angular velocity, and the obstacle crossing task having the maximum value. The questioning method is used for cognitive interference, which requires the participation of attentional resources. In early childhood, running patterns are not yet mature. When thinking occupies the attentional resources of running, unnatural running characteristics will appear, such as reduced running speed and reduced movement range. This response is the human body’s adjustment strategy to unstable environments [16]. Different from cognitive tasks, obstacle crossing is the interference of the external environment on the human body [17]. To avoid obstacles, children increase flexion joint angles and angular velocities at the moment of crossing to promote task completion. In Figure 3, for obstacle crossing tasks, most joint angles and angular velocities are greater than normal and cognitive tasks, which are reflected in the three joint ROM of lower limb, hip landing angle, hip pushing angle, knee landing angle, ankle landing angle, and angular velocities of hip and knee, which are still related to their task characteristics. There is a significant age difference in joint range of motion and joint angular velocity, and as age increases, there is a nonlinear decreasing trend in hip, knee, and ankle joint angles and angular velocity. Other research has shown that young children have a weaker ability to maintain physical stability during movement, and they often use rapid joint movement strategies for postural control. As age increases, joint movements become stable and coordinated [18]. Our results are consistent with the motor developmental characteristics of children’s movements.

Early childhood is a crucial stage for learning and mastering running skills, and some running characteristics may not even reach adult levels until mid-adolescence [19]. Task cost can explain the effect of interference tasks on running stability. Researchers often calculate task cost as an indicator in dual-task studies to describe the impact of additional loads on gait interference [20]. It is often expressed by calculating the task cost of the gait’s spatial parameters, and the commonly used characteristic parameters are speed and stride length. A positive value represents task cost, and the larger the value, the higher the task cost [21]. The theory of “Resource Sharing” points out that the human body’s attention resources are limited. When multiple tasks are performed simultaneously, they compete for resources. When the total amount of required resources exceeds the total supply, interference between tasks will occur, resulting in task costs [22]. The complex external environment can interfere with the stability of the human posture, requiring a series of motion control and postural responses to maintain a stable state [23]. Therefore, researchers have proposed the concept of “Priority Theory”. When a change in running mode is observed and task completion is not affected, it can be considered that the persons have chosen a “task priority” gait strategy. Different people make different choices. The elderly may be more inclined to maintain movement, which is “action first” running strategy, thereby ignoring task completion, whereas younger people tend to handle tasks [24,25], which is a “task first” strategy. In this study, we found that both cognitive and obstacle crossing tasks had task costs, and cognitive task costs were greater than those of obstacle crossing tasks. Children adopt a “task first” running strategy for different interference tasks. When facing cognitive tasks, their overall joint motion decreases and they use a strategy of reducing joint motion to promote cognitive task completion. When facing obstacle crossing tasks, due to the characteristics of the task itself, children increase joint motion to cope with interference.

Interference is ubiquitous in daily life. The postural control system of early childhood is in the developmental stage. For exceptional children, postural control ability will affect their normal life. Therefore, research on the dual-task of normal children can help children with special diseases formulate appropriate exercise prescriptions and promote the improvement of their living standards. From the full text data, both types of interference have an impact on children’s postural control, and the interference of cognitive dual-task is greater than obstacle crossing task. Children’s kinematic indicators show a nonlinear development trend, which means the level of children’s motor development at this stage is not stable; there is a certain degree of repetition, but it tends to develop upward on the whole. Therefore, special attention should be paid to the development characteristics of basic motor skills at this stage, and motor development methods such as whole sequence method and partial sequence method can be used for teaching motor skills. For children in this age group, using games is in line with their physical and mental development characteristics, and the interference environment can be set in the game intervention programs to strengthen the exercise of postural control.

## 5. Conclusions

Running characteristics of early childhood are affected by interference tasks and age. Under interference tasks, the overall characteristics of running in children aged 3–8 show an increasing trend in running cycle time and a decreasing trend in stride length, step length, cadence, and speed. Both cognitive and obstacle crossing tasks have task costs, and cognitive task costs are greater than obstacle crossing. Children adopt a “task first” running strategy when facing interference tasks. When facing cognitive tasks, their overall joint motion decreases and they use a strategy of reducing joint motions to promote cognitive task completion. When facing obstacle crossing tasks, due to the characteristics of the task itself, children increase joint motion to cope with interference. In terms of age, the running characteristics show a nonlinear development trend in various indicators, with a certain degree of recurrence and high variability in adjacent age groups. The dual-task interference paradigm of “postural-cognition” can be used as a tool to promote the development of basic motor skills in early childhood.

### 5.1. Limitations

This study had the following limitations: (1) we only tested the joint angle index of the sagittal axis, lack the angle analysis of the coronal and vertical axes; (2) we only conducted the test of running kinematics indicators, and lacked the test and calculation of internal indicators; and (3) gender characteristics of children were not analyzed.

### 5.2. Research Prospect

In the future, the joint angle can be comprehensively analyzed, and the angle analysis of coronal axis and vertical axis can be studied. In order to gain a deeper understanding of the interference mechanism of postural control in early childhood, research on neuromuscular regulation, such as electromyography analysis, muscle strength calculation, etc., can be increased by increasing the analysis of gender characteristics of postural control in early childhood and setting more interference types to understand interference effect on postural control in children.

## Figures and Tables

**Figure 1 sensors-24-01534-f001:**
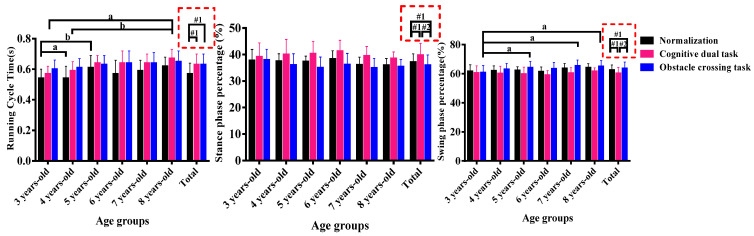
The temporal characteristics of running kinematics. Notes: #1, compares with the normalization group; #2, compares with the cognitive dual-task (*p* < 0.05); a, compares with 3-year-old group; b, compares with 4-year-old group (*p* < 0.05).

**Figure 2 sensors-24-01534-f002:**
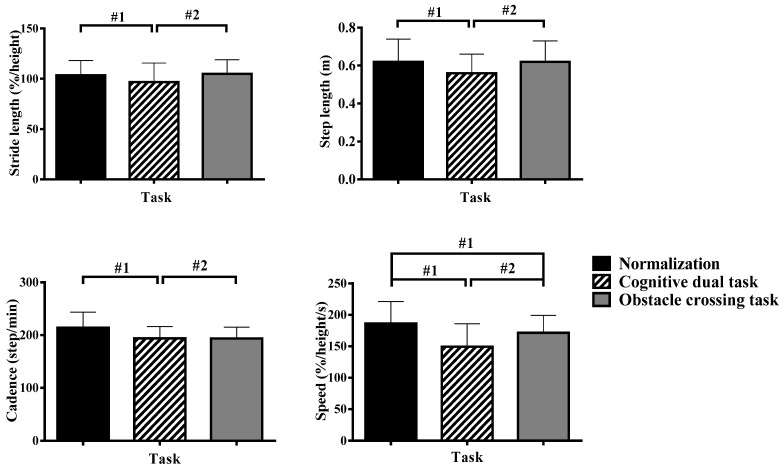
Running spatial characteristics of different tasks. Notes: #1, compares with the normalization group; #2, compares with the cognitive dual-task (*p* < 0.05).

**Figure 3 sensors-24-01534-f003:**
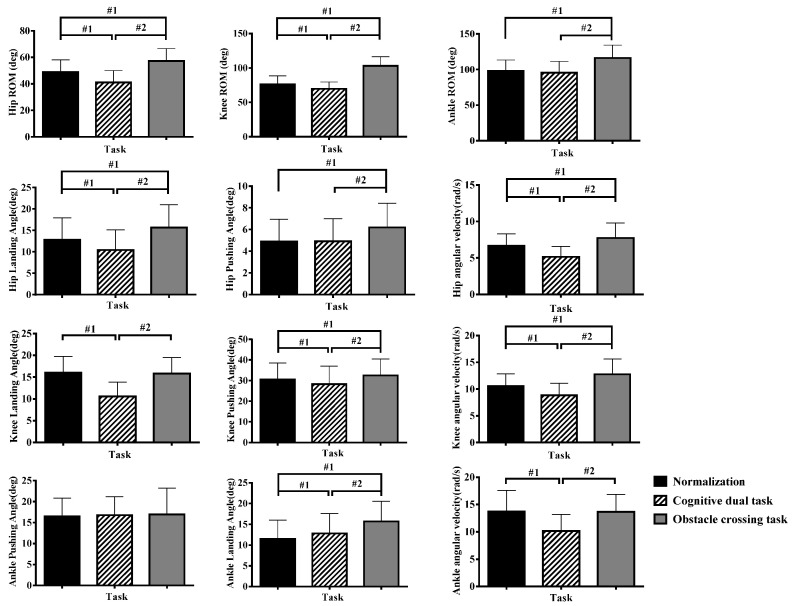
Joint kinematic characteristics of different task. Notes: #1, compares with the normalization group; #2 compares with the cognitive dual-task (*p* < 0.05).

**Figure 4 sensors-24-01534-f004:**
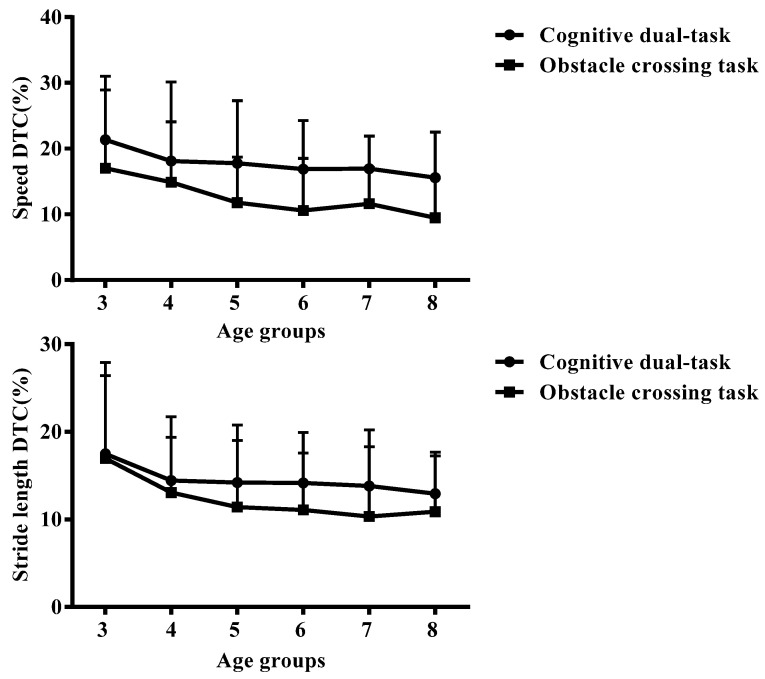
Age trend of dual-task costs.

**Table 1 sensors-24-01534-t001:** Kinematic parameters and definition in running.

Parameters	Definition
Running cycle time (s)	The time between one foot landing and the foot landing again
Stance phase percentage (%)	Stance phase times as a percentage of total running cycle time
Swing phase percentage (%)	Swing phase times as a percentage of total running cycle time
Stride length (%/height)	The longitudinal distance between one foot landing and landing again
Step length (m)	The longitudinal distance between one foot and the other foot
Cadence (step/min)	The number of steps run in 1 min
Speed (%/height/s)	The distance from one foot to the other divided by the running cycle time
Hip range of motion (deg)	The maximum hip flexion angle minus the minimum flexion angle during the running cycle
Knee range of motion	The maximum knee flexion angle minus the minimum flexion angle during the running cycle
Ankle range of motion	The maximum ankle flexion angle minus the minimum flexion angle during the running cycle
Hip landing angle (deg)	Hip flexion angle at landing moment
Hip pushing angle (deg)	Hip extension angle at pushing moment
Hip angular velocity (rad/s)	Maximum angular velocity of hip during running cycle
Knee landing angle (deg)	Knee flexion angle at landing moment
Knee pushing angle (deg)	Knee extension angle at pushing moment
Knee angular velocity (rad/s)	Maximum angular velocity of knee during running cycle
Ankle landing angle (deg)	Ankle plantar flexion angle at landing moment
Ankle pushing angle (deg)	Ankle dorsi flexion angle at pushing moment
Ankle angular velocity (rad/s)	Maximum angular velocity of ankle during running cycle

**Table 2 sensors-24-01534-t002:** Basic Information of Participants.

Age Group	*n*	Age (Years)	Height (cm)	Weight (kg)	BMI (kg/m^2^)
3 (3 ≤ χ < 4)	30	3.80 ± 0.15	103.30 ± 4.70	17.46 ± 2.38	16.37 ± 2.09
4 (4 ≤ χ < 5)	30	4.53 ± 0.31	108.78 ± 4.39 ^a^	18.89 ± 2.01	15.95 ± 1.34
5 (5 ≤ χ < 6)	30	5.53 ± 0.32	116.28 ± 5.08 ^ab^	21.70 ± 3.18 ^a^	16.02 ± 1.94
6 (6 ≤ χ < 7)	30	6.60 ± 0.32	122.93 ± 4.60 ^abc^	25.58 ± 4.43 ^abc^	16.88 ± 2.36
7 (7 ≤ χ < 8)	30	7.52 ± 0.26	126.33 ± 6.03 ^abc^	26.49 ± 6.51 ^abc^	16.43 ± 2.82
8 (8 ≤ χ < 9)	30	8.40 ± 0.23	132.65 ± 5.23 ^abcde^	32.39 ± 6.87 ^abcde^	18.34 ± 3.54 ^abc^
Overall	180	6.05 ± 1.64	118.31 ± 11.30	23.73 ± 6.83	16.67 ± 2.54

Notes: ^a^, compares with 3-year-old group; ^b^, compares with 4-year-old group; ^c^, compares with 5-year-old group; ^d^, compares with 6-year-old group; ^e^, compares with 7-year-old group (*p* < 0.05).

**Table 3 sensors-24-01534-t003:** Multiple-comparison ANOVA of running spatiotemporal parameters.

Parameter	Running Task	3-Year-Old	4-Year-Old	5-Year-Old	6-Year-Old	7-Year-Old	8-Year-Old
Stride length (%/height)	Normalization	99.25 ± 11.99	99.89 ± 13.14	101.92 ± 7.42	99.12 ± 11.99	107.52 ± 18.76	112.70 ± 17.09 ^abcd^
	Cognitive dual-task	94.87 ± 17.46	94.56 ± 16.77	92.80 ± 5.94	95.62 ± 8.53	100.13 ± 13.26	102.86 ± 8.35 ^cd^
	Obstacle crossing	99.95 ± 13.96	106.24 ± 11.01	104.80 ± 9.66	105.56 ± 17.11	104.69 ± 16.18	107.93 ± 15.49
Step length (m)	Normalization	0.54 ± 0.07	0.54 ± 0.08	0.61 ± 0.12	0.63 ± 0.09 ^ab^	0.66 ± 0.12 ^ab^	0.72 ± 0.12 ^abcd^
	Cognitive dual-task	0.50 ± 0.10	0.51 ± 0.10	0.54 ± 0.11	0.57 ± 0.07	0.60 ± 0.09 ^ab^	0.62 ± 0.07 ^abc^
	Obstacle crossing	0.52 ± 0.07	0.55 ± 0.09	0.62 ± 0.08	0.67 ± 0.08	0.66 ± 0.08	0.70 ± 0.11
Cadence (step/min)	Normalization	225.45 ± 22.04	226.52 ± 31.92	214.49 ± 28.12	214.17 ± 34.06	209.30 ± 27.75	193.76 ± 16.93 ^abc^
	Cognitive dual-task	210.84 ± 19.20	205.49 ± 21.09	187.43 ± 16.25 ^ab^	190.89 ± 24.75 ^a^	188.40 ± 17.68 ^ab^	179.84 ± 17.70 ^ab^
	Obstacle crossing	203.72 ± 18.14	199.53 ± 18.15	188.91 ± 19.51	192.16 ± 22.06	190.55 ± 24.65	186.66 ± 21.35 ^a^
Speed (%/height/s)	Normalization	200.69 ± 41.63	189.45 ± 41.30	182.66 ± 24.22	180.72 ± 28.17	184.97 ± 36.30	178.69 ± 32.23
	Cognitive dual-task	174.39 ± 42.22	160.61 ± 36.08	149.75 ± 30.93	132.32 ± 30.64 ^a^	142.64 ± 34.03 ^a^	136.42 ± 25.97 ^a^
	Obstacle crossing	173.82 ± 32.50	175.19 ± 27.63	173.80 ± 24.67	168.48 ± 27.12	168.16 ± 25.04	170.06 ± 29.11

Notes: ^a^, compares with 3-year-old group; ^b^, compares with 4-year-old group; ^c^, compares with 5-year-old group; ^d^, compares with 6-year-old group (*p* < 0.05).

**Table 4 sensors-24-01534-t004:** Multiple-comparison ANOVA of joint kinematic parameters.

Parameter	Running Task	3-Year-Old	4-Year-Old	5-Year-Old	6-Year-Old	7-Year-Old	8-Year-Old
Hip ROM (deg)	Normalization	48.96 ± 9.41	51.01 ± 9.64	50.89 ± 7.44	46.56 ± 9.00	49.36 ± 10.79	46.45 ± 9.00
	Cognitive dual-task	46.59 ± 11.13	41.99 ± 10.20	42.72 ± 7.17	39.08 ± 8.31	38.78 ± 6.93 ^a^	37.35 ± 6.69 ^a^
	Obstacle crossing	58.98 ± 11.84	59.34 ± 10.69	59.62 ± 8.93	57.84 ± 5.77	54.55 ± 9.28	53.30 ± 6.88
Knee ROM (deg)	Normalization	77.75 ± 12.00	75.00 ± 12.37	77.98 ± 9.27	72.72 ± 9.33	76.93 ± 16.73	76.31 ± 14.18
	Cognitive dual-task	74.55 ± 9.16	69.91 ± 13.76	70.93 ± 8.56	67.42 ± 8.31	67.89 ± 8.24	67.17 ± 9.33
	Obstacle crossing	104.81 ± 7.55	106.48 ± 14.47	108.25 ± 14.71	98.77 ± 10.72	102.42 ± 14.33	97.76 ± 13.72 ^c^
Ankle ROM (deg)	Normalization	105.80 ± 22.27	97.67 ± 14.03	93.67 ± 20.05	97.09 ± 14.85	97.92 ± 4.98	94.51 ± 6.64
	Cognitive dual-task	106.25 ± 16.31	94.44 ± 14.85	91.46 ± 11.51 ^a^	91.04 ± 21.22 ^a^	96.80 ± 7.39	93.72 ± 16.15
	Obstacle crossing	120.49 ± 19.62	118.00 ± 15.43	122.37 ± 16.34	113.55 ± 16.46	112.13 ± 19.18	110.79 ± 18.78
Hip Landing Angle (deg)	Normalization	12.37 ± 5.01	13.19 ± 5.48	11.69 ± 6.0	14.03 ± 5.03	13.83 ± 4.68	11.60 ± 4.48
	Cognitive dual-task	12.26 ± 4.99	10.99 ± 4.99	9.20 ± 4.80	10.75 ± 4.44	10.64 ± 4.23	8.62 ± 4.40 ^a^
	Obstacle crossing	14.92 ± 4.50	17.14 ± 6.96	15.17 ± 4.98	16.85 ± 5.29	16.47 ± 5.41	13.31 ± 4.24
Hip Pushing Angle (deg)	Normalization	5.80 ± 2.18	4.36 ± 2.03	5.36 ± 1.80	4.44 ± 1.71	4.29 ± 2.19	5.10 ± 2.03
	Cognitive dual-task	5.87 ± 2.10	5.20 ± 2.62	4.65 ± 1.77	4.19 ± 2.05 ^a^	4.78 ± 1.51	4.79 ± 2.14
	Obstacle crossing	6.99 ± 4.86	7.33 ± 5.14	6.13 ± 4.00	6.73 ± 4.01	4.46 ± 3.63	5.50 ± 3.10
Knee Landing Angle (deg)	Normalization	16.34 ± 3.21	17.33 ± 3.62	13.85 ± 2.49 ^a^	16.29 ± 2.89 ^a^	16.79 ± 2.49 ^c^	15.49 ± 3.13
	Cognitive dual-task	13.16 ± 1.81	12.77 ± 3.36	8.30 ± 2.29 ^ab^	9.98 ± 2.83 ^ab^	8.99 ± 2.85 ^ab^	10.30 ± 3.41 ^a^
	Obstacle crossing	15.13 ± 6.83	17.37 ± 8.22	15.35 ± 8.58	15.81 ± 6.94	16.60 ± 2.78	14.73 ± 5.33
Knee Pushing Angle (deg)	Normalization	29.14 ± 8.37	29.45 ± 8.39	31.79 ± 7.72	31.83 ± 7.16	30.13 ± 9.08	30.43 ± 8.37
	Cognitive dual-task	27.83 ± 6.66	28.68 ± 10.38	28.46 ± 8.17	29.31 ± 8.53	28.16 ± 9.14	27.20 ± 9.81
	Obstacle crossing	34.57 ± 8.10	34.54 ± 7.98	33.93 ± 6.10	32.29 ± 8.56	30.64 ± 6.81	29.27 ± 9.33
Ankle Landing Angle (deg)	Normalization	8.16 ± 2.15	12.05 ± 3.82 ^a^	12.80 ± 3.69 ^a^	13.15 ± 3.27 ^a^	12.90 ± 3.82 ^a^	11.87 ± 4.78 ^a^
	Cognitive dual-task	11.40 ± 3.24 ^#1^	11.24 ± 4.22	10.37 ± 4.86 ^#1^	14.21 ± 4.04 ^#1abc^	14.42 ± 3.68 ^c^	13.85 ± 3.81 ^c^
	Obstacle crossing	10.58 ± 3.14	15.09 ± 5.75 ^#1#2^	15.04 ± 5.77 ^#2^	17.28 ± 4.53 ^#2a^	15.68 ± 5.18 ^#1^	17.09 ± 5.01 ^#1#2a^
Ankle Pushing Angle (deg)	Normalization	14.79 ± 3.37	15.60 ± 4.14	15.65 ± 2.90	16.40 ± 4.13 ^a^	18.60 ± 4.67 ^a^	18.00 ± 5.17 ^ac^
	Cognitive dual-task	15.07 ± 3.64	16.18 ± 3.40	16.96 ± 3.14	17.06 ± 3.68	18.88 ± 6.19	16.54 ± 4.93
	Obstacle crossing	13.88 ± 3.83	16.04 ± 5.80	15.76 ± 4.82	17.65 ± 6.93	19.12 ± 7.12	19.17 ± 6.99
Hip angular velocity (rad/s)	Normalization	6.94 ± 1.83	7.30 ± 1.67	6.89 ± 1.23	6.34 ± 1.47	6.48 ± 1.35	6.11 ± 1.82
	Cognitive dual-task	6.07 ± 1.56	5.53 ± 1.88	5.29 ± 1.09	4.65 ± 1.14 ^a^	4.83 ± 1.06 ^a^	4.46 ± 1.30 ^a^
	Obstacle crossing	8.19 ± 2.43	8.51 ± 2.33	7.86 ± 1.66	7.46 ± 1.64	7.46 ± 2.34	6.98 ± 1.66
Knee angular velocity (rad/s)	Normalization	11.98 ± 2.30	11.98 ± 2.52	10.78 ± 1.82	9.51 ± 1.89 ^ab^	9.69 ± 1.95 ^ab^	9.30 ± 1.52 ^abc^
	Cognitive dual-task	10.80 ± 1.98	9.88 ± 2.98	9.01 ± 1.35 ^a^	8.03 ± 1.58 ^a^	7.90 ± 1.72 ^ab^	7.38 ± 1.25 ^abc^
	Obstacle crossing	14.64 ± 3.15	13.43 ± 2.22	12.65 ± 1.13 ^a^	12.28 ± 2.07 ^a^	12.32 ± 4.57 ^a^	11.20 ± 1.63 ^abc^
Ankle angular velocity (rad/s)	Normalization	15.82 ± 3.36	14.82 ± 3.98	14.76 ± 3.76	12.26 ± 3.18	12.79 ± 3.77	11.82 ± 3.56
	Cognitive dual-task	12.38 ± 3.15	10.62 ± 3.21	10.20 ± 2.76	9.96 ± 2.13 ^a^	9.51 ± 2.89 ^a^	8.28 ± 2.35 ^ab^
	Obstacle crossing	14.03 ± 2.96	13.42 ± 3.15	13.75 ± 2.27	13.05 ± 3.03	14.36 ± 4.07	13.34 ± 3.51

Notes: ^a^, compares with 3-year-old group; ^b^, compares with 4-year-old group; ^c^, compares with 5-year-old group; ^#1^, compares with the normalization group; ^#2^ compares with the cognitive dual-task (*p* < 0.05).

**Table 5 sensors-24-01534-t005:** Multiple-comparison ANOVA of task cost in different running task.

Age Group	*n*	Speed		Stride Length	
		Cognitive Dual-Task (%)	Obstacle Crossing Task (%)	Cognitive Dual-Task (%)	Obstacle Crossing Task (%)
3 years-old	30	21.35 ± 9.65	17.01 ± 11.89	17.49 ± 8.93	16.95 ± 10.97
4 years-old	32	18.11 ± 12.04	14.89 ± 9.20	14.45 ± 4.92	13.06 ± 8.66
5 years-old	30	17.80 ± 9.52	11.79 ± 6.92	14.22 ± 6.55	11.41 ± 7.61
6 years-old	30	16.88 ± 7.40	10.59 ± 7.92	14.16 ± 5.76	11.09 ± 6.47
7 years-old	30	16.94 ± 4.98	11.61 ± 5.54	13.81 ± 6.39	10.34 ± 7.95 ^a^
8 years-old	30	15.59 ± 6.91	9.48 ± 6.07 ^a^	12.95 ± 4.29	10.88 ± 6.81
Overall	180	17.78 ± 8.82 *	12.59 ± 8.51	14.51 ± 6.37 *	12.30 ± 8.41

Notes: *, means a significant difference between two tasks (*p* < 0.05); ^a^, compares with 3-year-old group (*p* < 0.05).

## Data Availability

The original contributions presented in the study are included in the article/Appendix A, further inquiries can be directed to the corresponding author.

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
