# Peer review of "Effects of Task Interference on Kinematics and Dual-Task Cost of Running in Early Childhood"

_sensors, 2024, doi:10.3390/s24051534_

Round 1
Reviewer 1 Report
Comments and Suggestions for Authors
The authors should be commended by the amount of work conducted for the manuscript. Please find my comments below.
Introduction
1. L39-40: This sentence is difficult to understand, suggest to re-word.
Method
1. L94: authors' need to clarify the meaning of data, what does it contain?
2. L95: it is recommended to add a reference to justify the use of Davis model, which is typically used for assessing gait performance in children (https://pubmed.ncbi.nlm.nih.gov/19321345/)
Results
1. L143, Figure 1 shows the effect of tasks and age on running cycle, rather than other way round. Please correct.
2. It is not clear how running cycle is defined.
3. The y-axis title for Figure 1a should be Running Cycle Time, rather than running cycle, as the unit is in seconds.
4. The caption for Figure 1 does not seem to be appropriate.
5. L148-L150: it is not clear what parameter the authors' tying investigate
6. It is recommended to use consistent terminology, such as stance phase percentage, swing phase percentage.
7. L159-160: it is not clear what does the authors meant by main and interactive effect.
8. Figure 2: it is not clear whether the results were averaged across all participants?
9. Figure 3: hip/knee/ankle landing and pushing need to be defined.
10. L190: There is no angular acceleration presented in any results?
11. Author seems to be quite focused on trends, e.g. L148, L151, L184, L189, L190 and L247. However, the trends were not cleared reflected in any of the bar charts or table. It is recommended add a few typical plots, e.g. line graphs to how clearly the trends. A good example would be Figure 2 in this paper: https://journals.sagepub.com/doi/10.1177/107110070402501215?url_ver=Z39.88-2003&rfr_id=ori:rid:crossref.org&rfr_dat=cr_pub%20%200pubmed
Discussion
1. L244-245: need to be more explicit about the meaning of joint angle, e.g. joint ROM, plantar/dorsi flexion (ankle), flexion/extension (knee and hip).
2. A follow-on question, which joint(s) showed prominent changes? is it ankle, knee or hip or combinations?
Comments on the Quality of English LanguageModerate English editing is required.
Author Response
----------------------------------------------------------------------------------------
Reply to the Review Report of Sensors-2865442-Panchao Zhao
-----------------------------------------------------------------------------------------
We are very grateful to the Review for the helpful comments and useful suggestions on our manuscript Sensors-2865442. We have read this review carefully and have revised our manuscript according to the review’s comments. Revised or/and added parts are marked with the red text in the manuscript. The follows are our reply item by item:
Introduction
1. Comment: L39-40: This sentence is difficult to understand, suggest to re-word.
Reply: Thanks for this reminding and helpful suggestion. The sentence has been modified and highlighted in red in the content of Introduction.
Method
1. Comment: L94: authors' need to clarify the meaning of data, what does it contain?
Reply: Thanks for this reminding and helpful suggestion. The corresponding table (seen in Table 1) has been added to the paper to explain the test data and its definition.
2. Comment: L95: it is recommended to add a reference to justify the use of Davis model, which is typically used for assessing gait performance in children (https://pubmed.ncbi.nlm.nih.gov/19321345/).
Reply: Thanks for this reminding and helpful suggestion. The corresponding references have been added here, seen in References [8].
Results
1. Comment: L143, Figure 1 shows the effect of tasks and age on running cycle, rather than other way round. Please correct.
Reply: Thanks for this reminding and helpful suggestion. The sentence has been modified, and the relevant sentences in the text have been modified, and the text has been marked red.
2. Comment: It is not clear how running cycle is defined.
Reply: The running cycle has been defined, as shown in Table 1.
3. Comment: The y-axis title for Figure 1a should be Running Cycle Time, rather than running cycle, as the unit is in seconds.
Reply: Thanks for this reminding and helpful suggestion. The y-axis of Figure 1 a has been changed to: Running Cycle Time, as shown in Figure 1 a.
4. Comment: The caption for Figure 1 does not seem to be appropriate.
Reply: Thanks for this reminding and helpful suggestion. The title of Figure 1 has been changed to: The temporary characteristics of running kinematics.
5. Comment: L148-L150: it is not clear what parameter the authors' tying investigate.
Reply: Thanks for this reminding and helpful suggestion. The indicators and their meanings have been explained in Figure 1, and the text at 148-150 has been modified.
6. Comment: It is recommended to use consistent terminology, such as stance phase percentage, swing phase percentage.
Reply: Thanks for this reminding and helpful suggestion. All the terms in the text have been unified.
7. Comment: L159-160: it is not clear what does the authors meant by main and interactive effect.
Reply: Thanks for this reminding and helpful suggestion. The explanations for the main and interactive effects have been added to the content under 2.1 design.
8. Comment: Figure 2: it is not clear whether the results were averaged across all participants?
Reply: Thanks for this reminding and helpful suggestion. The result is the average of all participants, which has been added in the 2.5 Data processing paragraph.
9. Comment: Figure 3: hip/knee/ankle landing and pushing need to be defined.
Reply: Thanks for this reminding and helpful suggestion. The landing and pushing angles of the lower limb joints have been defined, seen in Table 1.
10. Comment: L190: There is no angular acceleration presented in any results?
Reply: The data results of angular acceleration are shown in Figure 3 and Table 4, and their chart titles have been modified.
11. Comment: Author seems to be quite focused on trends, e.g. L148, L151, L184, L189, L190 and L247. However, the trends were not cleared reflected in any of the bar charts or table. It is recommended add a few typical plots, e.g. line graphs to how clearly the trends. A good example would be Figure 2 in this paper: https://journals.sagepub.com/doi/10.1177/107110070402501215?url_ver=Z39.88-2003&rfr_id=ori:rid:crossref.org&rfr_dat=cr_pub%20%200pubmed
Reply: Thanks for this reminding and helpful suggestion. The age development of kinematics shows a non-linear trend, therefore, kinematic data is presented in a tabular form. Dual task cost is a comprehensive indicator with a linear development trend, which is more evident in the image. Therefore, a line chart of task cost data has been added in the last part of 3.5 Dual task cost.
Discussion
1. Comment: L244-245: need to be more explicit about the meaning of joint angle, e.g. joint ROM, plantar/dorsi flexion (ankle), flexion/extension (knee and hip).
Reply: Thanks for this reminding and helpful suggestion. The specific meanings of joint angle and angular velocity are shown in Table 1, and this sentence also adds specific joint angle explanations.
2. Comment: A follow-on question, which joint(s) showed prominent changes? is it ankle, knee or hip or combinations?
Reply: Thanks for this reminding and helpful suggestion. Relevant text has been added to the fourth paragraph of the discussion and highlighted in red.
Thank you again for your time in reviewing our manuscript.
Best wishes for you.
Sincerely yours,
Panchao Zhao, Kai Ma, Zhongqiu Ji, and Guiping Jiang
Feb 21, 2024.

Reviewer 2 Report
Comments and Suggestions for Authors
The authors provide an analysis of task interference kinematics and dual task cost in early childhood during running. This is important as potential screening activities for abnormal gait or postural deficiencies with running may be better informed by these data. While these analyses provide important insights into development of running and kinematics during this developmental period of childhood, there are several clarifications that are required to explore these concepts more effectively:
1) Line 66 – “ANOVA” has not yet been introduced as an abbreviation; this must be spelled out when first mentioned.
2) Line 73 – what is meant by “did not meet the requirements”? It would be helpful to specify exactly what data were deleted and for what reason(s). It appears 20 of the 200 participants did not meet specifications for inclusion in analyses (based on Table 1). Please clarify the reason for exclusion of the 20 individuals’ data. This could be included early in the Results section.
3) Line 87 – it is mentioned that Master’s and Doctoral students collected these data, but it is not specified how many individuals collected these data. Were there any safeguards in place to ensure standardized and consistent data collection? What was the training for these multiple individuals?
4) Lines 113-114 – it would be helpful to provide additional information regarding the assessment of the cognitive level of the children. How exactly was this assessed? What criteria were in place to ensure adequate cognitive ability?
5) When discussing the calculation of “dual task cost,” it would be helpful to further provide a high level overview of how DTC is interpreted – this is not a common metric for those outside of the joint/gait kinematic space.
6) Was there randomization of the order of running tasks (i.e., cognitive dual tasks and obstacle crossing running test) following familiarization/normalized running test? If not, what is the impact of a potential learning effect (if all participants performed the obstacle test last, as an example)? This should be listed as a significant limitation if randomization of tasks did not occur.
7) A key area of improvement needed in the Discussion section is an explanation of how these findings can impact decisions or guidance when working with children in this age range, to promote optimal gait development. Your introduction includes the statement “… to enable timely screening of children with motor disorders…” – your manuscript would be strengthened by confirming how these findings would do that.
8) What are needed “next steps” for this line of research? How should the field take these findings and expand on them with follow-up research to address the limitations you’ve identified?
Author Response
----------------------------------------------------------------------------------------
Reply to the Review Report of Sensors-2865442-Panchao Zhao
-----------------------------------------------------------------------------------------
We are very grateful to the Review for the helpful comments and useful suggestions on our manuscript Sensors-2865442. We have read this review carefully and have revised our manuscript according to the review’s comments. Revised or/and added parts are marked with the red text in the manuscript. The follows are our reply item by item:
1. Comment: Line 66 – “ANOVA” has not yet been introduced as an abbreviation; this must be spelled out when first mentioned.
Reply: Thanks for this reminding and helpful suggestion. The full spelling has been added in Line 66.
2. Comment:Line 73 – what is meant by “did not meet the requirements”? It would be helpful to specify exactly what data were deleted and for what reason(s). It appears 20 of the 200 participants did not meet specifications for inclusion in analyses (based on Table 1). Please clarify the reason for exclusion of the 20 individuals’ data. This could be included early in the Results section.
Reply: Thanks for this reminding and helpful suggestion. 73 lines have been restated and specific exclusions and shedding criteria have been added.
3. Comment: Line 87 – it is mentioned that Master’s and Doctoral students collected these data, but it is not specified how many individuals collected these data. Were there any safeguards in place to ensure standardized and consistent data collection? What was the training for these multiple individuals?
Reply: Thanks for this reminding and helpful suggestion. Part of the content has been added and modified. It is expounded in the last part of 2.3participation, which explains everyone's responsibilities clearly.
4. Comment: Lines 113-114 – it would be helpful to provide additional information regarding the assessment of the cognitive level of the children. How exactly was this assessed? What criteria were in place to ensure adequate cognitive ability?
Reply: Thanks for this reminding and helpful suggestion. Before the test, by reviewing the learning materials of children of different age groups and discussing with the head teacher, we can understand the cognitive development level of children and formulate cognitive problems for children of different age groups. Added in the text and highlighted in red, please refer to the red highlighted content in 2.4.2 Procedures and Protocol for details.
5. Comment: When discussing the calculation of “dual task cost,” it would be helpful to further provide a high level overview of how DTC is interpreted – this is not a common metric for those outside of the joint/gait kinematic space.
Reply: Thanks for this reminding and helpful suggestion. A relevant overview of task costs has been added to the last paragraph of the discussion, and a reference has been added, see Reference 20.
6. Comment: Was there randomization of the order of running tasks (i.e., cognitive dual tasks and obstacle crossing running test) following familiarization/normalized running test? If not, what is the impact of a potential learning effect (if all participants performed the obstacle test last, as an example)? This should be listed as a significant limitation if randomization of tasks did not occur.
Reply: Yes, after the normal test, the test of cognitive dual task and obstacle crossing task is carried out randomly. After one child is tested, the test of the next child is carried out. The questions of cognitive task are set according to different age groups, while the height of obstacles in crossing task is different according to the height of each person. In 2.4.2 Procedures and Protocol, the title content subscript is red.
7. Comment: A key area of improvement needed in the Discussion section is an explanation of how these findings can impact decisions or guidance when working with children in this age range, to promote optimal gait development. Your introduction includes the statement “… to enable timely screening of children with motor disorders…” – your manuscript would be strengthened by confirming how these findings would do that.
Reply: Thanks for this reminding and helpful suggestion. A paragraph has been added at the end of the discussion, suggesting some decisions and guidance.
8. Comment: What are needed “next steps” for this line of research? How should the field take these findings and expand on them with follow-up research to address the limitations you’ve identified?
Reply: Thanks for this reminding and helpful suggestion. The limitation has been modified and a research Prospect section has been added. Please refer to the Research Prospect for specific details.
Thank you again for your time in reviewing our manuscript.
Best wishes for you.
Sincerely yours,
Panchao Zhao, Kai Ma, Zhongqiu Ji, and Guiping Jiang
Feb 21, 2024.

Round 2
Reviewer 1 Report
Comments and Suggestions for Authors
Thank you very much for the revisions and I am happy to confirm that all of my concerns have been resolved.
Comments on the Quality of English LanguageMinor English edits are needed.